# A rapid calprotectin test for the diagnosis of pleural effusion

Pedro Casado-Rey[1], Lorena Vázquez-Iglesias[2,3]*, Maribel Botana-Rial[4], María Amalia Andrade-Olivié[1], Lucía Ferreiro-Fernández[5], Esther San José Capilla[6], Ana Nuñez-Ares[7], Elena Bollo de Miguel[8], Virginia Pajares-Ruíz[9], Alberto Fernández-Villar[3]

1 Clinical Chemistry Department, Hospital Álvaro Cunqueiro, EOXI Vigo, Spain, 2 CINBIO, Universidade de Vigo, Vigo, Spain, 3 Grupo de Investigación en Química Coloidal, Instituto de Investigación Sanitaria Galicia Sur (IIS Galicia Sur), SERGAS-UVIGO, Vigo, Spain, 4 Pulmonary Department, Hospital Álvaro Cunqueiro, EOXI Vigo, PneumoVigoI+I Research Group, Health Research Institute Galicia Sur (IIS Galicia Sur), Vigo, Spain, 5 Pulmonary Department, University Hospital Complex of Santiago, EOXI Santiago, Spain, Health Research Institute Santiago (IDIS), Santiago de Compostela, Spain, 6 Clinical Chemistry Department, University Hospital Complex of Santiago, EOXI Santiago, Spain, 7 Pulmonary Department, University Hospital Complex of Albacete, Albacete, Spain, 8 Pulmonary Department, University Hospital Complex of León, León, Spain, 9 Pulmonary Department, Hospital Santa Creu i Sant Pau of Barcelona, Barcelona, Spain

* lorena.vazquez@uvigo.es

**Data Availability Statement:** All relevant data are within the paper and its Supporting Information files.

## Abstract

In previous studies, measuring the levels of calprotectin in patients with pleural effusion (PE) was an exceptionally accurate way to predict malignancy. Here, we evaluated a rapid method for the measurement of calprotectin levels as a useful parameter in the diagnosis of malignant pleural effusion (MPE) in order to minimise invasive diagnostic tests. Calprotectin levels were measured with Quantum Blue® sCAL (QB®sCAL) and compared with the gold standard reference ELISA method. Calprotectin levels in patients with benign pleural effusion (BPE) were significantly higher ($p < 0.0001$) than for MPE patients. We measured the sensitivity, specificity, positive predictive value (PPV), negative predictive value (NPV), and positive and negative likelihood ratios (LRs) for a cut-off value of $\leq 14,150$ ng/mL; the diagnostic accuracy was 64%. The odds ratio for PE calprotectin levels was 10.938 (95% CI [4.133 − 28.947]). The diagnostic performance of calprotectin concentration was better for predicting MPE compared to other individual parameters. Comparison of two assays showed a slope of 1.084, an intercept of 329.7, and a Pearson correlation coefficient of 0.798. The Bland–Altman test showed a positive bias for the QB®sCAL method compared to ELISA fCAL®. Clinical concordance between both these methods was 88.5% with a Cohen kappa index of 0.76 (95% CI [0.68 − 0.84]). We concluded that QB®sCAL is a fast, reliable, and non-invasive diagnostic tool for diagnosing MPE and represents an alternative to ELISA that could be implemented in medical emergencies.

## Introduction

There are multiple causes of pleural effusion (PE) [1, 2]; in cases of unilateral PE, the most frequent and important diagnosis that must be established or excluded is malignancy. The search for the cause of PE almost always involves an analysis of the pleural fluid (PF) obtained by

**Funding:** Project PI13/01538 (FIS-FEDER), Xunta de Galicia (CN/2014), grant of societies AEER (2015), SOGAPAR (2014). Lorena Vázquez was supported by Fundación Biomédica Galicia Sur with In-tegrated research programs of SEPAR (PII Neumología Intervencionista 2015).

**Competing interests:** No authors have competing interests

thoracentesis [3]. In undiagnosed cases, additional invasive diagnostic tests must be indicated, even though these are sometimes unnecessary. Measurement of biomarkers in PF may represent a reliable tool that can enhance clinical diagnostic pathways. The incorporation of new molecular biomarkers into clinical practice is limited by the high cost of the testing equipment and the absence of biomarker validation in well-designed, prospective, multicentre studies [4].

In previous studies, we concluded that calprotectin levels measured in PE could predict malignancy with high degree of accuracy [5, 6]. However, these studies were limited, among other factors, by the small number of patients in which calprotectin was determined in our research laboratories. Thus, we carried out this multicentre study to validate calprotectin levels in PE as a diagnostic biomarker of malignancy in clinical settings [7]. In this study, calprotectin levels were measured using the conventional robotised equipment and employing the enzyme-linked immunosorbent assay (ELISA fCA®, Bühlmann Laboratories) routinely used in our laboratory [7, 8]. The results of this work were useful when making early clinical decisions about the treatment of these patients because the use of this marker can lead to better diagnoses and can help avoid possible adverse consequences. It is also important to highlight that the high morbidity and mortality associated with malignant PE means that a rapid diagnosis is required [9], and so these types of measurement procedures are not useful in medical emergencies [8, 10]. Therefore, we proposed the incorporation of calprotectin as a standard parameter in urgent diagnostic thoracentesis analyses to help guide the indication for other invasive procedures such as closed pleural biopsy or thoracoscopy. ELISA is currently considered the gold standard for the laboratory measurement of these kinds of parameters. However, using ELISA to measure calprotectin is a time consuming and labour-intensive way to complete just one test.

The use of predictive models that combine clinical data with biochemical parameters could improve the aetiological diagnosis of PE [11–14]. Indeed, Klimiuk et al. [11] proposed two predictive models with a remarkably high diagnostic accuracy to differentiate between tuberculous pleural effusion (TBPE) and non-TBPE. The first model included body temperature, white blood cell count, PF adenosine deaminase (ADA), and IP-10 levels. The second model was based on age, sex, body temperature, white blood cell count, PF lymphocyte percentage, and IP-10 levels. In relation to cancer, Valdés et al. [14] proposed a strategy for detecting malignant pleural effusion (MPE) by using different prognosis models that combine clinical, radiological, and analytical variables. The model which combines clinical-radiological criteria (absence of chest pain, fever, and compatible radiological images) and analytical criteria (carcinoembryonic antigen [CEA], neuron-specific enolase [NSE], serum cytokeratin 19 fragment [Cyfra 21.1], and the tissue-specific polypeptide antigen [TPS]), was the most accurate model. However, the diagnostic accuracy of a simple model, in which only the CEA is determined and assessed alongside clinical-radiological criteria, was remarkably similar and would be easier to implement in healthcare practice.

Thus, the objective of this study was to evaluate a rapid method to measure calprotectin levels in PF samples in urgent situations in order to differentiate between benign pleural effusion (BPE) and MPE, and to compare it with the results obtained from the gold standard ELISA method. We then proposed a decision algorithm for the urgent diagnosis of PE which included the measurement of calprotectin levels in PF together with other clinical variables with the aim of reducing the number of unnecessary invasive diagnostic tests carried out. We decided to perform a prospective multicentre study to evaluate a rapid monotest method, Quantum Blue® sCAL (QB®sCAL), to measure calprotectin levels as an urgent parameter for the diagnosis of MPE. This assay is based on lateral flow technology and is not time consuming. Given that ELISA fCAL® is the only calprotectin measurement method that has been evaluated for

use with PF to date [8], to confirm the clinical validity of QB®sCAL we correlated its results with those from the ELISA fCAL® method used in our previous multicentre study [7].

## Methods

### Study population

This was a prospective multicentre study that included five different hospitals in Spain: Hospital Álvaro Cunqueiro (EOXI Vigo), the University Hospital Complex of Santiago, University Hospital Complex of Albacete, University Hospital Complex of León, and Hospital Santa Creu i Sant Pau of Barcelona. Patients were recruited for this work between January 2014 and July 2017 and 307 patients with PE exudates and a specific diagnosis were included. All the centres applied the same protocol and only adults aged over 18 years were eligible for this study. The exclusion criteria were a previous diagnosis of pleurodesis; current treatment with intrapleural or systemic anti-neoplastic agents; or the presence of pus in the pleural space (empyema).

The PEs were diagnosed as previously described in the guidelines published by different medical societies [5, 6], as described in previous studies by our group [7]. The PEs were categorised as MPE, BPE, TBPE, parapneumonic PE (PNPE), or as a benign non-infective PE. The malignant origin of PEs was defined when malignant cells were identified upon cytological or histological examination or in a biopsy specimen.

### Demographic, clinical, radiological, and biochemical variables

We recorded the following demographic patient variables: age, gender, smoking status, previous cancer diagnosis, and clinical variables including chest pain, dyspnoea, cough, fever, and constitutional symptoms. Characteristic radiological images were classified as large and massive PE on chest X-rays. PF was collected with syringes without anticoagulant by thoracocentesis or during pleural biopsy before starting any treatment. The PF was then centrifuged at $800 \times g$ for 15 minutes and were immediately frozen in 0.5 mL aliquots at $-80°C$ until sample delivery on dry ice. The biochemical parameters of the PF were determined by each participant's centre before the beginning of any treatment by determining the following parameters: differential cell counts, pH, proteins, lactate dehydrogenase (LDH), glucose, and ADA. All the samples were included in the CHUVI Biobank (RETIC-FIS-ISCIII RD09/0076/00011).

### Ethical approval and consent to participation

The patient data and samples were obtained in full compliance with the clinical and ethical practices of the Spanish Government and the Declaration of Helsinki and the study protocol was approved by the Galicia Ethics Committee (2014/053). All the patients received written and oral information prior to their inclusion in the study and provided written informed consent before its commencement. Participant anonymity was maintained in all cases.

### Calprotectin determination

Calprotectin levels were measured with a Quantum Blue® sCAL quantitative Lateral Flow Assay (Bühlmann, Schönenbuch, Switzerland) and the sample quantification was carried out strictly respecting the manufacturer's instructions. The kit was validated for serum calprotectin and the assay was adapted for PF. Briefly, PF samples were diluted 1:100 and loaded into the test cartridge. After 12 min of incubation at room temperature, the signal intensities were measured quantitatively in a Quantum Blue Reader. According to the manufacturer's analytical characteristics: the lower and upper limits of quantification were 5 μg/mL (5,000 ng/mL) and 100 μg/mL (100,000 ng/mL), respectively.

## Statistical analysis

Quantitative data are presented as the median (25th–75th percentile) and as percentages and absolute frequencies for qualitative variables. The ability of calprotectin to discriminate between MPE and BPE was evaluated by calculating receiver operating characteristic curves (ROC), as well as the sensitivity, specificity, likelihood ratios (LRs) and their confidence intervals (CIs), positive predictive value (PPV), negative predictive value (NPV), and diagnostic accuracy. The cut-off point was the value that rendered the highest accuracy. Univariate logistic regression was performed to evaluate the ability of calprotectin alone to predict the diagnosis of MPE. This relationship was also examined for other demographic, clinical, radiological, and biochemical variables.

Unadjusted odds ratios (OR) were calculated as an estimate of the relative risk, and the corresponding 95% CIs were reported. Significant predictors in the univariate analysis ($p < 0.1$) were entered into a multivariate logistic regression model to assess the independent predictive value of calprotectin levels. A diagnostic algorithm derived from multivariate logistic regression analyses was developed to diagnose MPE, considering $p$ values $< 0.05$ as statistically significant. All the statistical analyses were performed with Statistical Package for Social Sciences software (version 21, IBM Corp., Armonk, NY).

## Assay validation

A total of 253 patients from centres 1 and 2 were included. PE calprotectin levels were measured using both QB®sCAL and ELISA fCAL®. The measurement range of the assays were 5,000 to 100,000 ng/mL for QB®sCAL and 400 to 24,000 ng/ml for ELISA fCAL®. Because of the large difference in the linearity of both these tests following optimisation for use with PF, these methods were analytically compared in the 48 samples whose results fell within the overlapping measurement range of both assays (5,000–24,000 ng/mL). Pearson linear correlation, coefficient of determination, and Passing–Bablok regression statistics were calculated for the linearity study and Bland–Altman plots were used to analyse the agreement between both assays. Methval software (Method Validator v1.19) was employed for data analyses.

Clinical concordance between the two methods was assessed in all 253 patient samples by categorising calprotectin values as positive or negative for malignancy according to the reference range of each assay and calculating the overall concordance and Cohen κ [15]. Positive diagnostic values were considered according to the calprotectin cut-off calculated in this study for QB®sCAL and as $\leq 6{,}323.2$ ng/ml for ELISA fCAL® [7]. Values higher than these were considered negative. Clinical concordance between the two tests was assessed by calculating the kappa index [16].

## Results

### Multicentre trial

A total of 307 consecutive patients were included in this study; 198 (64.5%) were diagnosed as BPE and 109 (35.5%) as MPE. Table 1 shows the characteristics of the study patients according to their classification as BPE or MPE. The demographic, clinical, radiological, and biochemical variables are shown in Table 2.

Table 3 presents the PE calprotectin concentrations according to the aetiology of the PE; the calprotectin level for BPE patients was significantly higher ($p < 0.0001$) than that of the MPE patients. Among the BPE groups, higher levels of calprotectin were found in PNPE or TBPE relative to non-infective PE. No significant differences were found between the MPE groups.

**Table 1. Aetiology of the pleural effusion.**

| Aetiology PE | All Patients *N* (%) |
|---|---|
| **Benign PE** | 198 (64.5) |
| **Tuberculous PE** | 21 (10.6) |
| **Parapneumonic PE** | 83 (41.9) |
| **Non-infective PE** | 94 (47.5) |
| **Malignant PE** | 109 (35.5) |
| **Lung cancer** | 59 (54.1) |
| **Mesothelioma** | 8 (7.3) |
| **Ovarian cancer** | 11 (10.1) |
| **Gastric cancer** | 5 (4.6) |
| **Breast cancer** | 14 (12.8) |
| **Haematological cancers** | 8 (7.3) |
| Others* | 4 (3.7) |

PE = pleural effusion

* 1 melanoma (centre 1), 1 bladder cancer (centre 1), 1 hepatocellular carcinoma (centre 2), and 1 metastatic cancer of unknown origin (centre 1).

In previous studies [5, 8] we evaluated the diagnostic accuracy of calprotectin measured by ELISA fCAL® to determine its utility in the differentiation of the PE origin (S1 Table). In this study, we analysed the diagnostic accuracy of calprotectin when using the QB®sCAL method (Table 4). The ROC curve analysis of calprotectin for MPE diagnosis was an AUC of 0.787 (95% CI [0.737 − 0.836]). For a cut-off value ≤ 14,150 ng/mL, the sensitivity, specificity, PPV, NPV, and positive and negative LRs, were 93.6%, 50%, 49%, 93%, 1.87, and 0.13, respectively, and the diagnostic accuracy was 64%.

Logistic regression was performed with a cut-off of ≤ 14,150 ng/mL (Table 5). Univariate logistic regression demonstrated a strong association between the low levels of calprotectin and malignancy, with a high unadjusted OR (14,571, $p = 0.000$). Univariate analysis also revealed significant associations between MPE and female sex ($p = 0.007$), massive PE in chest radiographs ($p = 0.000$), dyspnoea ($p = 0.000$), constitutional syndrome ($p = 0.000$), and the absence of fever ($p = 0.009$). The significant predictors were entered into a multivariate logistic regression model, which indicated that the association between female sex, massive PE, constitutional syndrome, absence of fever, and calprotectin levels ≤ 14,150 ng/mL remained significant predictors. The odds ratio for PE calprotectin levels ≤ 14,150 ng/mL was 10.938 (95% CI [4.133 − 28.947]). Thus, the diagnostic performance of calprotectin concentration in the prediction of MPE was better than that of other individual parameters.

Thus, as in our previous study [7], we proposed an algorithm for the diagnostic management of suspected MPE which included the determination of calprotectin with QB®sCAL (Fig 1).

## Assay validation

PE calprotectin levels measured by QB®sCAL and ELISA fCAL® assays are shown in Table 6. The analytical correlation obtained between these two methods was moderate with a Pearson correlation coefficient (*r*) of 0.798 and a determination coefficient ($R^2$) of 0.623. The intercept of the linear regression, calculated using Passing Bablok regression analysis (Fig 2A), was 329.7 (95% CI [−2,605.2 − 27,56.3], thereby significantly deviating from 0. Regarding the slope of the regression equation (1.084, 95% CI [0.878–1.553]), there was no significant deviation from 1.

**Table 2. Results of the clinical and analytical variables.**

| Clinical variables | BPE | MPE |
|---|---|---|
| Gender (male/female) [a] | 138/60 (69.7/30.3) | 59/50 (54.1/45.9) |
| Age (years)[b] | 65.5 (51–77.25) | 70 (60–80.5) |
| Previous cancer[a] | 35 (17.7) | 34 (31.2) |
| Smoker [a] | 106 (53.5) | 57 (52.3) |
| Dyspnoea [a] | 123 (62.1) | 89 (81.7) |
| Chest pain [a] | 97 (49) | 39 (35.8) |
| Cough [a] | 79 (30.9) | 35 (32.1) |
| Constitutional syndrome [a] | 28 (14.1) | 46 (42.2) |
| Haemoptysis [a] | 7 (3.5) | 5 (4.6) |
| Fever [a] | 72 (36.4) | 8 (7.3) |
| Massive effusion [a, c] | 17 (8.6) | 30 (27.5) |
| Analytical variables [b] | BPE | MPE |
| LDH (U/L) | 438 (283–998) | 522 (341–832) |
| Protein (g/dL) | 4.59 (3.9–5.03) | 4.4 (3.78–4.87) |
| pH | 7.39 (7.29–7.44) | 7.40 (7.32–7.45) |
| ADA (U/L) | 24 (19.4–40.62) | 20 (14.8–25.85) |
| Glucose (mg/dL) | 99.5 (76–127) | 105 (90–126) |
| Polymorphonuclear leukocytes (%) | 21 (5–59) | 9 (2–20) |
| Mononuclear leukocytes (%) | 70 (31.25–90) | 82 (50.5–94.25) |

BPE = benign pleural effusion; MPE = malignant pleural effusion; constitutional syndrome includes at least one of the following variables: asthenia, anorexia, and/or weight loss; LDH = lactate dehydrogenase; ADA = adenosine deaminase.

[a] Data are presented as frequencies and percentages

[b] Data are presented as medians and interquartile ranges (IQRs)

[c] Radiological effusion features: pleural effusion causing opacification of an entire hemithorax or when the fluid reaches the aortic arch.

The equation of the linear regression line was QB®sCAL = 1.084 * ELISA fCAL® + 329.7. The intercept and the slope included 0 and 1, respectively.

Bland-Altman tests were performed to assess bias across the measurement range. The results of the QB®sCAL method were an average 2,100 units higher than those obtained by the ELISA fCAL® method (Fig 2B). For analytical comparison, measurement ranges below 5,000 ng/mL obtained by the QB®sCAL method and above 24,000 ng/mL from the ELISA fCAL® method were excluded because of the lack of an exact calprotectin value.

The clinical concordance between the two methods was 88.5% (224/253) with a Cohen kappa index of 0.76 (95% CI [0.68 – 0.84]; Table 7). In 138 samples, the results were positive for these two methods; 86 samples presented concentrations above the specific cut-off point for each method (negative), 20 samples were positive by QB®sCAL and negative by ELISA fCAL®, and 9 were positive only by ELISA fCAL®.

## Discussion

Differentiating between MPE and BPE has prognostic and therapeutic implications. PF cytology has traditionally been the analytical method of choice for the detection of tumour cells. However, PF cytology is far from perfect for diagnosing MPE. First, the diagnostic sensitivity of LP cytology ranges from approximately 60% in metastatic malignancies to less than 30% in mesothelioma. Second, sensitivity is at least in part dependent on the experience of the

**Table 3. Calprotectin concentration in pleural fluid according to the type of pleural effusion.**

| Causes of PE | N (%) | Calprotectin (ng/mL) [a] | p-value |
|---|---|---|---|
| Benign PE [P] | 198 (64.5) | 14,150 (5,000–62,375) | |
| Tuberculous PE | 21 (10.6) | 27,100 (9,025–50,550) | |
| Parapneumonic PE | 83 (41.9) | 42,900 (10,100–100,000) | |
| Non-infective PE | 94 (47.8) | 6,300 (5,000–19,700) | |
| | | | **p < 0.0001** [P] |
| Malignant PE [P] | 109 (35.5) | 5,000 (5,000–5,850) | |
| Lung cancer | 59 (54.1) | 5,000 (5,000–5,700) | |
| Mesothelioma | 8 (7.3) | 5,000 (5,000–15,550) | |
| Ovarian cancer | 11 (10.1) | 5,000 (5,000–6,800) | |
| Gastric cancer | 5 (4.6) | 5,000 (5,000–12,250) | |
| Breast cancer | 14 (12.8) | 5,000 (5,000–6,375) | |
| Haematologic cancers | 8 (7.3) | 5,000 (5,000–5,000) | |
| Others [*] | 4 (3.7) | 5,000 (5,000–5,675) | |

PE = pleural effusion.

[a] Data are presented as the median and interquartile ranges.

[P] Pleural fluid calprotectin concentration in BPE compared to MPE, p < 0.0001.

[*] 1 melanoma, 1 bladder cancer, 1 hepatocellular carcinoma, and 1 metastatic cancer of unknown origin.

cytologist, tumour load, and amount of PF submitted. Another limitation is that standard cytology examination based on cytomorphology is often unable to distinguish between different types of malignancy (e.g., between adenocarcinoma and mesothelioma) without the use of special studies such as complementary immunohistochemistry in the diagnosis of malignant PEs. Nonetheless, patient cytology in conjunction with immunohistochemistry studies are insufficient to establish a diagnosis, thus requiring additional tests, usually in the form of invasive procedures to sample the pleura for histological examination. Therefore, new non-invasive biomarkers that improve the diagnostic sensitivity of cytology are required.

In previous studies using ELISA methods, we demonstrated that calprotectin is a useful diagnostic biomarker in the diagnosis of MPE [5, 7]. The current gold standard technique used for evaluation of calprotectin is the ELISA method, although chemiluminescent immunoassays (CLIA) and lateral flow immunochromatography techniques are also widely used [17]. ELISA methods have certain limitations including the fact that they are time consuming, tedious, and laborious, and they require specialised equipment and trained personal. These limitations generate turnaround delays and reduce the clinical utility in the diagnosis of MPE in clinical practice. Although faster than ELISA, CLIA calprotectin determination methods are

**Table 4. Diagnostic accuracy of pleural fluid calprotectin levels according to several cut-off values.**

| Cut-off[*] | S (%) | E (%) | PPV (%) | NPV (%) | LR⁺ | LR⁻ | DE (%) |
|---|---|---|---|---|---|---|---|
| ≤ 9,750 | 88.1 (80.7–92.9) | 59.1 (52.1–65.7) | 54.2 (46.9–61.4) | 90 (83.6–94.1) | 2.15 (1.80–2.58) | 0.20 (0.12–0.34) | 69.4 (64.0–74.3) |
| ≤ 14,150 | 93.6 (87.3–96.9) | 50 (43.1–56.9) | 50.7 (43.9–57.6) | 93.4 (87–96.8) | 1.87 (1.61–2.17) | 0.13 (0.06–0.027) | 65.5 (60.0–70.6) |
| ≤ **23,800** | 99.1 (95.0–99.8) | 39.9 (33.3–46.8) | 47.6 (41.2–54.1) | 98.8 (93.3–99.8) | 1.65 (1.47–1.85) | 0.02 (0.00–0.16) | 60.9 (55.4–66.2) |
| ≤ 24,850 | 100 (96.6–100) | 39.4 (32.9–46.3) | 47.6 (41.2–54.1) | 100 (95.3–100) | 1.65 (1.47–1.85) | 0 | 60.9 (55.4–66.2) |

Data are presented as the 95% confidence intervals.

[*] Calprotectin levels are expressed in ng/mL. S = sensitivity; E = specificity; PPV = positive predictive value, NPV = negative predictive value, LR⁺ = positive likelihood ratio; LR⁻ = negative likelihood ratio; DE = diagnostic efficiency.

**Table 5. Frequency variables and results from the univariate and multivariate logistic regression analyses.**

| Variable | MPE* | BPE* | Univariate analysis OR (95% CI) | p | Multivariate analysis OR (95% CI) | p |
|---|---|---|---|---|---|---|
| **Gender = female** | 50/109 (45.9) | 60/198 (30.3) | 1.949 (1.202–3.161) | 0.007 | 3.205 (1.596–6.435) | 0.001 |
| Age ≥ 67.5** | 61/108 (56.5) | 92/198 (46.5) | 1.495 (0.933–2.397) | 0.095 | 0.996 (0.524–1.894) | 0.991 |
| **Tobacco use** | 57/108 (52.8) | 106/196 (54.1) | 0.949 (0.593–1.520) | 0.827 | - | - |
| **Previous cancer** | 34/109 (31.2) | 35/192 (18.2) | 2.034 (1.178–3.512) | 0.011 | 1.492 (0.720–3.095) | 0.282 |
| Massive effusion [a] | 30/109 (27.5) | 17/198 (8.6) | 4.043 (2.109–7.753) | 0.000 | 4.088 (1.604–10.414) | 0.003 |
| PF calprotectin ≤ 14,150** | 102/109 (936) | 99/198 (50) | 14.571 (6.451–32.916) | 0.000 | 10.938 (4.133–28.947) | 0.000 |
| PMN > 50%*** | 7/109 (6.4) | 57/198 (28.8) | 0.170 (0.074–0.387) | 0.000 | 0.429 (0.146–1.263) | 0.125 |
| pH ≥ 7.40** | 58/107 (53.2) | 92/190 (96) | 1.261 (0.784–2.027) | 0.339 | - | - |
| LDH ≥ 477** | 59/107 (55.1) | 92/195 (47.2) | 1.376 (0.857–2.209) | 0.186 | - | - |
| ADA ≤ 23** | 67/105 (63.8) | 83/194 (42.8) | 2.358 (1.446–3.846) | 0.01 | 1.653 (0.867–3.152) | 0.127 |
| Glucose ≥ 102** | 61/109 (56) | 92/192 (47.2) | 1.381 (0.861–2.215) | 0.180 | - | - |
| **Dyspnoea** | 89/108 (82.4) | 123/198 (62.1) | 2.856 (1.611–5.063) | 0.000 | 1.482 (0.709–3.098) | 0.295 |
| **Pain chest** | 39/108 (36.1) | 97/197 (49.2) | 0.583 (0.360–0.943) | 0.028 | 1.065 (0.546–2.076) | 0.853 |
| **Cough** | 35/108 (32.4) | 79/198 (39.9) | 0.722 (0.441–1.183) | 0.196 | - | - |
| Constitutional syndrome [b] | 46/108 (42.6) | 28/197 (14,1) | 4.478 (2.577–7.782) | 0.000 | 4.175 (1.996–8.732) | 0.000 |
| **Haemoptysis** | 5/108 (4.6) | 7/197 (3.6) | 1.318 (0.408–4.256) | 0.645 | - | - |
| **Fever** | 8/108 (7.4) | 72/197 (36.5) | 0.139 (0.064–0.302) | 0.000 | 0.290 (0.114–0.736) | 0.009 |

BPE = benign pleural effusion; MPE = malignant pleural effusion; ADA = adenosine deaminase; LDH = lactate dehydrogenase; OR = odds ratio; CI = confidence interval; PMN = polymorphonuclear leukocytes.

* Data are presented as the frequencies and percentages

** median cut off

*** Clinical definition.

[a] Radiological effusion features: pleural effusion causing opacification of an entire hemithorax or when the fluid reaches the aortic arch

[b] Constitutional syndrome includes at least one of the following variables: asthenia, anorexia, and/or weight loss.

dependent on the availability of autoanalysers and, in this context, calprotectin determination in PF is a low-demand test. Thus, the ELISA and CLIA methods are not ideal for calprotectin determination in PF for the diagnosis of PE in most clinical laboratories.

The new lateral flow immunochromatographic tests are designed for use as a point-of-care testing (POCT) assay with individual samples, are economical, and can provide quick answers to urgent clinical questions [18, 19]. This is the first study to evaluate calprotectin levels in PE with the Quantum Blue® lateral flow immunochromatographic method in a multicentre cohort. This is an easy-to-use method that produces more rapid results than the ELISA methodology. The advantage of QB®sCAL compared to other POCT assays is that it correlates well with the ELISA fCAL® [20–24]. Indeed, our group used this ELISA fCAL® method in previous studies [7] and validated it for the determination of calprotectin in PF [8].

In the first part of this current work we assessed the diagnostic accuracy of calprotectin for the diagnosis of MPE, as measured with the QB®sCAL method in a multicentre study. A multivariate logistic regression demonstrated that calprotectin levels represent a more effective diagnostic procedure in medical emergencies compared to existing biomarkers. In addition, the integration of calprotectin and clinical and radiological variables can provide a promising approach to differential diagnosis between MPE and BPE during routine clinical practice.

In the second part, we evaluated the correlation of the QB®sCAL and ELISA fCAL® tests, with the analytical comparison results being poorer than expected. While the coefficients of correlation with a similar number of samples ($n = 26–54$) indicated by the manufacturer (Bühlmann Laboratories AG), ranged from 0.86 to 0.94 for faeces [20–22] and 0.94 for serum

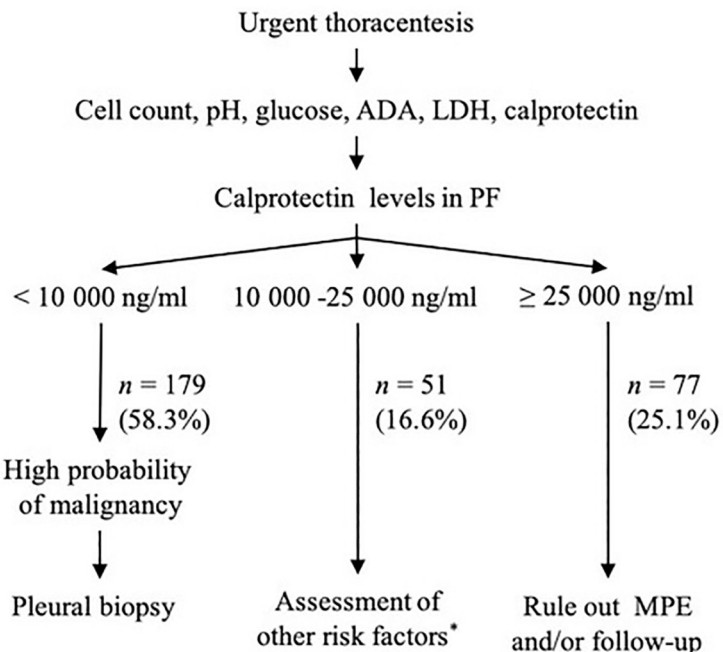

Urgent thoracentesis

Cell count, pH, glucose, ADA, LDH, calprotectin

Calprotectin levels in PF

< 10 000 ng/ml    10 000 -25 000 ng/ml    ≥ 25 000 ng/ml

$n = 179$ (58.3%)    $n = 51$ (16.6%)    $n = 77$ (25.1%)

High probability of malignancy    Assessment of other risk factors*    Rule out MPE and/or follow-up

Pleural biopsy

* Pleural biopsy indication when:
woman and/or massive effusion and/or constitutional syndrome
and/or absence of fever

**Fig 1. New clinical decision algorithm proposed for the urgent diagnosis of MPE using the QB®sCAL immunochromatographic method for determining calprotectin.** ADA = adenosine deaminase; LDH = lactate dehydrogenase; PF = pleural fluid; $N$ = frequency; MPE = malignant pleural effusion.

[23], in our study the correlation coefficient was 0.623 for PF. Coorevits et al. [24], evaluated the correlation between these methods with a similar method of comparison but using greater number of faeces samples ($n$ = 142) and found an $R^2$ of 0.89 between both methods. Similarly, Burri et al. [25] evaluated the Bülhmann methods in ascitic liquid, obtaining a correlation coefficient of 0.873, very similar to our results for PF. Other studies comparing the Bühlmann Quantum Blue rapid test with an ELISA kit found correlation coefficients between 0.53 and 0.94 [26].

We found significant differences in measurement ranges between the QB®sCAL and the ELISA fCAL® using PF. Thus, we calculated different cut-off points for each method: 14,150

**Table 6. Results for calprotectin level measurements using the ELISA fCAL® method versus the QB®sCAL method.**

| PE aetiology | N (%) * | Methods** | |
| --- | --- | --- | --- |
| | | ELISA fCAL® | QB®sCAL |
| BPE | 160 (63.2) | 14,170 (4,209.75–24,000) | 17,900 (5,700–66,675) |
| MPE | 93 (36.8) | 1,942 (1,023–3,125) | 5,000 (5,000–5,400) |

PE = pleural effusion; BPE = benign pleural effusion; MPE = malignant pleural effusion.

*Data are presented as frequencies and percentages

**Data are presented as medians and interquartile ranges.

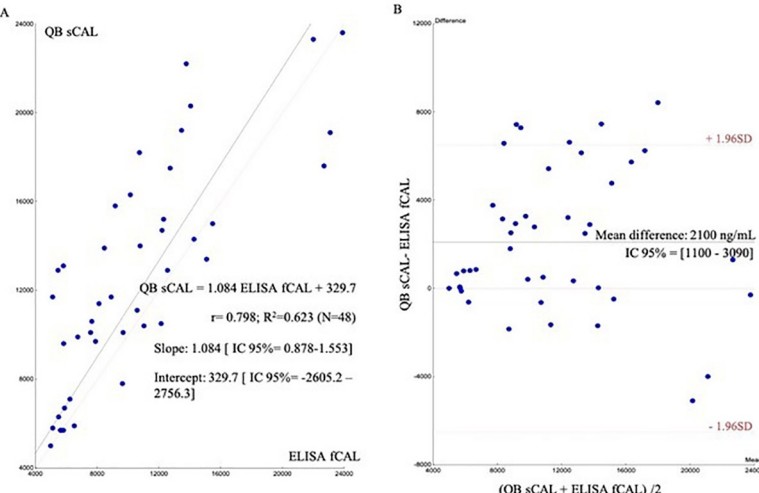

**Fig 2. Analytical comparison of QB®sCAL and ELISA fCAL®.** (a) Passing-Bablok linear regression of the results of calprotectin pleural fluid measurements for QB®SCAL and corresponding results for ELISA fCAL® performed in 48 patients (analytical correlation). (b) Bland-Altman difference plot for analysis of the differences between the two methods (analytical agreement).

ng/mL for QB®sCAL versus 6,233.2 ng/mL for ELISA fCAL® [7], which could explain our correlation data. The Bland-Altman plots clearly showed that the results offered by the POCT in PF were higher than those obtained by the ELISA fCAL®. In addition, the 95% CI of the mean difference did not include the value 0, and so we cannot be sure the absence of statistical differences between the two methods. This, therefore, means that the results of these two these analytical techniques were not interchangeable even though they were provided by the same manufacturer.

POCT is normally less precise, accurate, specific, or sensitive than testing performed in a clinical laboratory setting. In this study, the diagnostic accuracy of the calprotectin levels measured in PF by the QB®sCAL method were slightly lower than that of the ELISA fCAL® [7]. However, when the results were considered according to the cut-off points of each method, we observed that the percentage of agreement approached 90% with a considerable kappa index (0.76). This high degree of clinical agreement confirms the clinical validity of this biomarker for the diagnosis of MPE regardless of the procedure applied, as long as we use the appropriate cut-off points for each technique.

Calprotectin has been studied as a biomarker for several different diseases, applying POCT methodologies. Wouthuyzen-Bakker et al. [27], used calprotectin as a marker to diagnose

**Table 7. Clinical agreement between the results from the ELISA fCAL® and QB® sCAL methods[a].**

| ELISA fCAL® | QB®sCAL | | | |
|---|---|---|---|---|
| | Results | Positive** | Negative** | Total |
| | Positive* | 138 | 9 | 147 |
| | Negative* | 20 | 86 | 106 |
| | Total | 158 | 95 | 253 |

[a]**Cohen kappa index = 0.76 (95% CI [0.68 – 0.84])**

*Positive ≤ 6,233.2 ng/mL, negative > 6,233.2 ng/mL

**Positive ≤ 14,150 ng/mL, negative > 4,150 ng/mL.

chronic prosthetic joint infections, by measuring calprotectin in 52 consecutive patients using a lateral flow immunoassay. Applying a cut-off value of 50 mg/L, synovial calprotectin showed a sensitivity, specificity, positive predictive value, and negative predictive value of 86.7%, 91.7%, 81.3%, and 94.4%, respectively. Thus, measuring synovial calprotectin levels was a useful biomarker in the diagnostic work-up of patients with chronic pain, especially for the exclusion of prostatic joint infection prior to revision surgery. In turn, Lorenzo Drago et al. [28] analysed α-defensin POCT for diagnosis of prosthetic joint infections in synovial fluid. This study explains that the validation of the POCTs by laboratory professionals, evaluating the comparison and possible accordance with the reference enzyme immunoassay for the same analyte, is mandatory to achieve the diagnostic accuracy requirements necessary for routine use.

In another study, Burri et al. [25], evaluated the diagnostic capacity of measuring calprotectin in ascitic fluid for detecting a polymorphonuclear cell count $> 250/\mu L$ ascites. ELISA and a POCT lateral flow assay with the Quantum Blue® Reader (Bühlmann Laboratories) were used to measure calprotectin in ascitic fluid. These authors found that ascitic calprotectin could reliably predict polymorphonuclear count $> 250/\mu L$, which could prove useful in the diagnosis of spontaneous bacterial peritonitis, especially if a bedside testing device were readily available. Additionally, the correlation between ELISA and POCT was excellent ($r = 0.873$, $p < 0.001$).

Of note, the Quantum Blue Reader from Bühlmann Laboratories AG is a portable, connectable, and quantitative POCT which is based on immunochromatography [18]. Most POCTs are easy to use and are designed to process samples that do not require special handling. Thus, they do not require specialised staff and can be used outside clinical laboratories. However, one of the main limitations of this method is the preanalytical treatment of PF samples because prior to the analysis, they require centrifugation, supernatant separation, and dilution, all necessitating laboratory equipment which could limit the use of the Quantum Blue® sCAL method outside of emergency laboratories.

Our study confirms that calprotectin levels measured with a rapid quantitative lateral flow assay in combination with clinical variables, had excellent discriminatory values for predicting suspected MPE in medical emergencies. In conclusion, the rapid quantification of PF calprotectin using the QB®sCAL assay represents a fast and reliable method for diagnosing of MPE which could be used an alternative to the ELISA method. QB®sCAL achieves good accuracy in PE calprotectin measurement with a good discrimination between MPE and BPE patients. Therefore, QB®sCAL is a non-invasive diagnostic tool that should be implemented in clinical practice.

## Supporting information

**S1 Table. Diagnostic accuracy of the pleural fluid calprotectin levels measured by ELISA fCAL®.**
(DOCX)

**S1 Data.**
(XLSX)

**S2 Data.**
(SAV)

**S3 Data.**
(XLS)

## Author Contributions

**Conceptualization:** Pedro Casado-Rey, Lorena Vázquez-Iglesias, Maribel Botana-Rial, María Amalia Andrade-Olivié, Alberto Fernández-Villar.

**Data curation:** Pedro Casado-Rey, Lorena Vázquez-Iglesias, María Amalia Andrade-Olivié.

**Formal analysis:** Pedro Casado-Rey, Lorena Vázquez-Iglesias, Lucía Ferreiro-Fernández, Esther San José Capilla, Ana Nuñez-Ares, Elena Bollo de Miguel, Virginia Pajares-Ruíz.

**Investigation:** Lorena Vázquez-Iglesias, Maribel Botana-Rial, María Amalia Andrade-Olivié.

**Methodology:** Pedro Casado-Rey, Lorena Vázquez-Iglesias, Lucía Ferreiro-Fernández, Esther San José Capilla, Ana Nuñez-Ares, Elena Bollo de Miguel, Virginia Pajares-Ruíz.

**Supervision:** Maribel Botana-Rial, Alberto Fernández-Villar.

**Validation:** Lorena Vázquez-Iglesias, María Amalia Andrade-Olivié.

**Visualization:** Alberto Fernández-Villar.

**Writing – original draft:** Pedro Casado-Rey, Lorena Vázquez-Iglesias, Maribel Botana-Rial, María Amalia Andrade-Olivié, Alberto Fernández-Villar.

**Writing – review & editing:** Pedro Casado-Rey, Lorena Vázquez-Iglesias, Maribel Botana-Rial, María Amalia Andrade-Olivié, Alberto Fernández-Villar.

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
