## [Decision Letter · Decision Letter 0]

4 Mar 2021

PONE-D-21-01414

A rapid calprotectin test for the diagnosis of pleural effusion

PLOS ONE

Dear Dr. Vázquez-Iglesias,

Thank you for submitting your manuscript to PLOS ONE. After careful consideration, we feel that it has merit but does not fully meet PLOS ONE’s publication criteria as it currently stands. Therefore, we invite you to submit a revised version of the manuscript that addresses the points raised during the review process.

In adding to the issues raised by the reviewers, please clarify the main purpose of the paper.

Is a validation of using calprotectin in pleural fluid as a surrogate for leucocytes, in diagnosing malignant or infections effusions? Or is it a comparison between calprotectin POCT and ELISA methods.

Please explain abbreviations at first occurrence if used > 3times, remember the abstract is a separate unity.

The manuscript could benefit from native English editing.

We look forward to receiving your revised manuscript.

Kind regards,

Pal Bela Szecsi, M.D. D.M.Sci.

Academic Editor

PLOS ONE

Journal Requirements:

2) Your ethics statement should only appear in the Methods section of your manuscript. If your ethics statement is written in any section besides the Methods, please delete it from any other section.

Reviewers' comments:

Reviewer's Responses to Questions

**Comments to the Author**

1. Is the manuscript technically sound, and do the data support the conclusions?

Reviewer #1: Yes

Reviewer #2: Yes

2. Has the statistical analysis been performed appropriately and rigorously? 

Reviewer #1: Yes

Reviewer #2: Yes

3. Have the authors made all data underlying the findings in their manuscript fully available?

Reviewer #1: Yes

Reviewer #2: Yes

4. Is the manuscript presented in an intelligible fashion and written in standard English?

Reviewer #1: Yes

Reviewer #2: Yes

5. Review Comments to the Author

Reviewer #1: The study " A rapid calprotectin test for the diagnosis of pleural effusion" addresses to an important topic in clinical practice: the diagnosis of malignant pleural effusion, especially in cases where the patient has no previous diagnosis of malignant disease,

and/or the cytological examination of the pleural effusion was negative for cancer cells.

The article presents a clear Introduction and objectives and the methodology used is adequate for the study.

Comments:

1) In the pag 4 - Methods - the authors report that only "healthy adults over 18 years were eligible for the study..." If they have pleural effusion, they are not healthy!! I suggest correction.

2) In the page 10 - Results - table 3: The values of Calprotectin presented as 5000 ng/mL in MPE, does it mean that there was no detection with the assay used? If the lower detection value was considered for cases with Calprotectin undetectable by the assay used, this criterion did not prejudice the statistical analysis?

3) The authors should refer to the average cost of the test and the equipment, especially considering its use in developing countries;

4) In the study there is no reference to the positivity of the PE cytology. What was the criterion used in the definition of MPE: cytology or biopsy? It would be interesting to report on the performance of cytology in the case of MPE.

5) Considering the values obtained for specificity, PPV and diagnostic accuracy, would the authors really suggest using this POCT assay for clinical management ??

Reviewer #2: This paper proposes a rapid measurement of calprotectin to diagnose pleural effusion. The diagnostic performance of calprotectin level is verified in clinical practice, and the correlation between the measurement method proposed in this paper and the standard methods is analyzed, which provides effective guidance for clinical diagnosis. However, there are still some problems in this paper:

1. Some abbreviations in this paper are not given full names, so that the paper is less readable.

2. What is the main contribution of this paper? Is the calprotectin level measurement method proposed in this paper for the first time? If so, this paper should focus on the proposed measurement method. Or does the experiment prove that the method can be used in clinical practice? The contribution of this article should be clearly written.

3. The advantages of the Quantum Blue® sCAL measurement method are mentioned in the paper, but the disadvantages of this method are not clearly pointed out, which should be supplemented in this paper.

4. In the assay validation part, is it reasonable to take samples of 5000-24000 to analyze the correlation between the two measurement methods? Need to point out the reason for this analysis, or supplement the correlation analysis of the two measurement methods on 253 samples.

5. The paper does not give the diagnosis result of calprotectin level measured by ELISA method, and the result similar to Table 4 for ELISA method should be given.

6. Does the cut-off value in this paper have guiding significance for the follow-up clinical diagnosis of pleural effusion?

7. there are many relevant predictive models have been proposed for the diagnosis of pleural effusion, the authors should make a deep investigation on this issue in the introduction part, such as the works [1]-[4].

[1] J. M. Seixas, J. Faria, J. B. O. Souza Filho, A. F. M. Vieira, A. Kritski, and A. Trajman, "Artificial neural network models to support the diagnosis of pleural tuberculosis in adult patients," International Journal of Tuberculosis and Lung Disease, vol. 17, pp. 682-686, May 2013.

[2] C.-C. Shu, J.-Y. Wang, C.-L. Hsu, L.-T. Keng, K. Tsui, J.-F. Lin, et al., "Diagnostic role of inflammatory and anti-inflammatory cytokines and effector molecules of cytotoxic T lymphocytes in tuberculous pleural effusion," Respirology, vol. 20, pp. 147-154, Jan 2015.

[3] J. Klimiuk, A. Safianowska, R. Chazan, P. Korczynski, and R. Krenke, "Development and Evaluation of the New Predictive Models in Tuberculous Pleuritis," Adv Exp Med Biol, vol. 873, pp. 53-63, 2015.

[4] Li C, Hou L, Sharma B, Li H, Chen C, Li Y, Zhao X, Huang H, Cai Z, Chen H. Developing a new intelligent system for the diagnosis of tuberculous pleural effusion. Computer Methods & Programs in Biomedicine, 2018, 153: 211-225.

6. PLOS authors have the option to publish the peer review history of their article (what does this mean?). If published, this will include your full peer review and any attached files.

Reviewer #1: No

Reviewer #2: No

---

## [Author Response · Author response to Decision Letter 0]

9 Apr 2021

ANSWERS TO Reviewer 1

We would like to thank the comments about our work. We have considered the questions and comments raised by the referee and here we provide the itemized answers. Changes have been highlighted in red in a new version of the manuscript for an easier revision. 

Comment 1: In the pag 4 - Methods - the authors report that only "healthy adults over 18 years were eligible for the study..." If they have pleural effusion, they are not healthy!! I suggest correction.

Response 1: We apologize for the mistake and we corrected this sentence in the manuscript: Only adults over 18 years were eligible for this study. 

Comment 2: In the page 10 - Results - table 3: The values of Calprotectin presented as 5000 ng/mL in MPE, does it mean that there was no detection with the assay used? If the lower detection value was considered for cases with Calprotectin undetectable by the assay used, this criterion did not prejudice the statistical analysis?

Comment 2.1: was no detection with the assay used?

Response 2.1: As we commented in methods section (Assay validation), the measurement range of QB®sCAL is 5000 to 100000 ng/mL. This does not mean that lowers values are no detected. In this test, it is recommended to assign the values of the detection limits (upper or lower) to those samples that present values above or below the detection limits.

Comment 2.2: If the lower detection value was considered for cases with Calprotectin undetectable by the assay used, this criterion did not prejudice the statistical analysis?

Response 2.2: We are in agreement with the referee that this criterion could be affect the statistical analysis. In the data sheet of Quantum Blue® sCAL assay says: The results showed linearity within the indicated measuring range of 5000to 10000 ng/mL of the Quantum Blue® sCAL assay. For that reason, we decide to apply this criterion to do the analysis even though that the statistical analysis could be worse.

Comment 3: The authors should refer to the average cost of the test and the equipment, especially considering its use in developing countries;

Response 3: According to your request the average cost of calprotectin measurement by the Quantum Blue method has a cost of 20 euros per test. This price includes the cost of reagents and the loan of the equipment (Quantum Blue Reader) to the clinical laboratory by Bühlmann company. Initially, we have not included this data in the manuscript because we think that it does not contribute to the research, but if you consider that it is important we include it in the manuscript.

Comment 4: In the study there is no reference to the positivity of the PE cytology. What was the criterion used in the definition of MPE: cytology or biopsy? It would be interesting to report on the performance of cytology in the case of MPE.

Response 4: As suggested by the referee we add this sentence in methods section: The MPE was diagnosed by cytological or histologic examination. Malignant origin was defined when malignant cells were identified upon cytological or histological examination or in a biopsy specimen.

In the discussion section we add this comment: Pleural fluid cytology has traditionally been the analytical method of choice for the detection of tumor cells in pleural fluid. However, pleural fluid cytology is far from perfect for diagnosing MPE. First, the diagnostic sensitivity of pleural fluid cytology varies and is higher (up to 60%) in metastatic malignancies than in mesothelioma (~30%). Second, sensitivity is at least in part dependent on the experience of the cytologist, the tumor load and the amount of fluid submitted. Another limitation is that standard cytology examination based on cytomorphology is often unable to distinguish between different types of malignancy (e.g., between adenocarcinoma and mesothelioma) without the use of special studies such as immunohistochemistry. Immunohistochemistry complements cytology in the diagnosis of malignant pleural effusions. But the patient’s cytology in conjunction with immunohistochemistry. studies will not establish a diagnosis, and additional investigations will be required – usually invasive procedures to sample pleura for histological examination.

Comment 5: Considering the values obtained for specificity, PPV and diagnostic accuracy, would the authors really suggest using this POCT assay for clinical management??

Response 5: According to your request, we considering that this POCT assay could be uses for clinical management. The principal reason is that our study confirms that calprotectin levels measured with this POCT assay in combination with clinical variables, has an excellent discriminatory value for predicting suspected MPE in medical emergency. The QB®sCAL achieves good accuracy in PE calprotectin measurement with a good discrimination between MPE and BPE patients. For a proportion of patients with pleural effusions, pleural fluid cytology not establish a diagnosis, and additional investigations will be required, usually invasive procedures to sample pleura for histological examination. QB®sCAL is a noninvasive diagnostic tool that should be implemented in clinical practice. 

ANSWERS TO Reviewer 2

We would like to thank the comments about our work. We have considered the questions and comments raised by the referee and here we provide the itemized answers. Changes have been highlighted in red in a new version of the manuscript for an easier revision

Comment 1: Some abbreviations in this paper are not given full names, so that the paper is less readable.

Response 1: We apologize for the mistake of not referring the full names of some abbreviations. We have been corrected the errors. 

Comment 2: What is the main contribution of this paper? Is the calprotectin level measurement method proposed in this paper for the first time? If so, this paper should focus on the proposed measurement method. Or does the experiment prove that the method can be used in clinical practice? The contribution of this article should be clearly written.

Response 2: We apologize for the confusion regarding the main purpose of this manuscript. Both arguments are right. This is the first time that this calprotectin measurement method (QB®sCAL) is used in PF samples and also this study confirms that this method can be used in clinical practice. According to your request, we correct the objective to clarify the main purpose of the paper. We intend to mean that calprotectin is a complementary biomarker to the rest of the biochemistry parameters and that could help in the differential diagnosis of PE when we perform an urgent thoracentesis. At this time, the gold standard technique used for evaluation of calprotectin is the ELISA method, but this method cannot be used in urgent situations. We correct the main objective in the manuscript as follows: The objective of this study was to evaluate a rapid method to measure calprotectin levels in PF samples in urgent situations for differentiating between benign (BPE) and malignant pleural effusion (MPE) and compare it with the ELISA method.

Furthermore, the text in Discussion section has also been modified to explain the this better, as follows: Our study confirms that calprotectin levels measured with a rapid quantitative lateral flow assay in combination with clinical variables, has an excellent discriminatory value for predicting suspected MPE in medical emergency.

In conclusion, a rapid quantification of PF calprotectin using QB®sCAL assay represents a fast and reliable method for diagnosing of MPE and it could be an alternative to the ELISA method. The QB®sCAL achieves good accuracy in PE calprotectin measurement with a good discrimination between MPE and BPE patients. QB®sCAL is a non-invasive diagnostic tool that should be implemented in clinical practice.

Comment 3: The advantages of the Quantum Blue® sCAL measurement method are mentioned in the paper, but the disadvantages of this method are not clearly pointed out, which should be supplemented in this paper.

Response 3: As suggested by the referee, we have added the disadvantages of the method. We added this part in the manuscript: On the other hand, it is important to mention that a principal limitation of this method is the preanalytical treatment of PF samples because the samples require a centrifugation and dilution prior the analysis. This preanalytical treatment require adequate laboratory equipment that could limit the use of the Quantum Blue® sCAL method outside the emergency laboratories.

Comment 4: In the assay validation part, is it reasonable to take samples of 5000-24000 to analyze the correlation between the two measurement methods? Need to point out the reason for this analysis, or supplement the correlation analysis of the two measurement methods on 253 samples.

Response 4: As stated in the methodology section (Assay validation), due to the difference in the linearity range of both tests following optimization for use in PF, the analytical comparison of this methods was only possible made with the 48 samples that were included in the measurement range shared by both assays (5000-24000 ng/mL). However, the clinical concordance is independent of this linearity range of the methods used because of that all the samples (253) were included. As we explained, we calculated a cut-off point for each method and categorized the samples as positive and negative based on that cut-off point.

Comment 5. The paper does not give the diagnosis result of calprotectin level measured by ELISA method, and the result similar to Table 4 for ELISA method should be given.

Response 5: We agree with your comment and we add the diagnostic accuracy of pleural fluid calprotectin for ELISA method. 

Comment 6: Does the cut-off value in this paper have guiding significance for the follow-up clinical diagnosis of pleural effusion?

Response 6: 

This is a very interesting comment and it may be possible that the calprotectin values could be used in the follow-up of the pleural effusion diagnosis but unfortunately the calprotectin measure for this study only was made in the initial time. 

Comment 7: there are many relevant predictive models have been proposed for the diagnosis of pleural effusion, the authors should make a deep investigation on this issue in the introduction part, such as the works [1]- [4].

Response 7: According to your request, we revised these references and redacted the introduction section including them

---

## [Decision Letter · Decision Letter 1]

21 May 2021

A rapid calprotectin test for the diagnosis of pleural effusion

PONE-D-21-01414R1

Dear Dr. Vázquez-Iglesias,

We’re pleased to inform you that your manuscript has been judged scientifically suitable for publication and will be formally accepted for publication once it meets all outstanding technical requirements.

Kind regards,

Pal Bela Szecsi, M.D. D.M.Sci.

Academic Editor

PLOS ONE

Additional Editor Comments (optional):

Reviewers' comments:

Reviewer's Responses to Questions

**Comments to the Author**

1. If the authors have adequately addressed your comments raised in a previous round of review and you feel that this manuscript is now acceptable for publication, you may indicate that here to bypass the “Comments to the Author” section, enter your conflict of interest statement in the “Confidential to Editor” section, and submit your "Accept" recommendation.

Reviewer #1: (No Response)

Reviewer #2: All comments have been addressed

2. Is the manuscript technically sound, and do the data support the conclusions?

Reviewer #1: Yes

Reviewer #2: Yes

3. Has the statistical analysis been performed appropriately and rigorously? 

Reviewer #1: Yes

Reviewer #2: Yes

4. Have the authors made all data underlying the findings in their manuscript fully available?

Reviewer #1: Yes

Reviewer #2: Yes

5. Is the manuscript presented in an intelligible fashion and written in standard English?

Reviewer #1: Yes

Reviewer #2: Yes

6. Review Comments to the Author

Reviewer #1: The authors adequately answered the questions and included the relevant points of the comments in the new version of the text.

Reviewer #2: The authors have solved all the problems that I have raised, now the paper is ready for publication.

7. PLOS authors have the option to publish the peer review history of their article (what does this mean?). If published, this will include your full peer review and any attached files.

Reviewer #1: No

Reviewer #2: No

---

## [Editor Report · Acceptance letter]

2 Jun 2021

PONE-D-21-01414R1 

A rapid calprotectin test for the diagnosis of pleural effusion 

Dear Dr. Vázquez-Iglesias:

I'm pleased to inform you that your manuscript has been deemed suitable for publication in PLOS ONE. Congratulations! Your manuscript is now with our production department. 

Kind regards, 

on behalf of

Dr. Pal Bela Szecsi 

Academic Editor

PLOS ONE